# Automated Mining of Hinge-like Protein Modules from AlphaFold PAE: BCR-Parts

## Abstract

We introduce *BCRParts*, a simple pipeline that turns single-structure predictions into candidates for mechanically separable protein "parts". The method segments the AlphaFold PAE into $k \in \{2, 3\}$ blocks, scores block–contrast (BCR), and assigns significance by permutation $p$-values with Benjamini–Hochberg FDR applied *once per run*. On a curated cohort we processed 1,476 targets in parallel. The score distribution shows a clear right tail, and—importantly—the BCR-derived score correlates with an independent PAE diagnostic, the asymmetry index (Pearson $r = 0.59$, Spearman $\rho = 0.69$, Fig. 3). Using a two-sided calibration (see Methods), 314/1,476 rows pass BH at $q \leq 0.05$ (Fig. 2); top candidates and attributes (e.g., hinge length, opening angle) are summarized in Table 1. External evidence (PDBFlex/CoDNaS) could not be robustly retrieved in time for this submission; we therefore report coverage diagnostics (Table 2) and leave full cross-dataset validation to future work.

## 1 Introduction

Large language models (LLMs) and diffusion models are rapidly changing protein design. Recent systems generate backbones and sequences with controllable topology and function, and some designs have been validated experimentally [Watson et al., 2023, Ingraham et al., 2023]. Foundation models further unify sequence, structure, and function, enabling multi-modal conditioning and iterative design loops [Hayes et al., 2025]. In parallel, structure prediction now handles joint protein–nucleic acid–ligand complexes at high accuracy, closing the loop between generative proposals and interaction-aware screening [Abramson et al., 2024]. These advances motivate a new question at the interface of AI design and molecular robotics: can we automatically mine and standardize protein "parts" that behave like reusable robotic components (hinges, rods, latches), and feed them into modern generative pipelines as constraints and interfaces?

We argue that building such a parts library requires two ingredients. First, a data-driven modularity detector that flags contiguous subchains with strong intra-block coherence and weak inter-block coupling, using confidences intrinsic to large-scale predicted structures. Second, a mechanical interface layer that encodes how parts connect (e.g., termini geometry, axis orientation, and "latch-like" residue pairs) and how they interoperate with widely used bio-orthogonal connectors and switches.

This paper introduces *BCRParts*, an AI-first pipeline that (i) analyzes AlphaFold DB predictions to extract contiguous two-block (and $k$-block) modular candidates from predicted aligned error (PAE) heatmaps, (ii) quantifies block separability by a block contrast ratio (BCR) statistic with robust null models, and (iii) annotates mechanical and interface features to surface candidates as robotic parts. Concretely, we use a spectral bipartition on a PAE-derived similarity graph (Fiedler-vector sign and a contiguity heuristic) to propose block boundaries, then compute quantile- and mean-based BCR statistics and assess significance against rotation/permutation nulls with Benjamini–Hochberg

FDR control across cohorts [von Luxburg, 2007, Benjamini and Hochberg, 1995]. We rely on AlphaFold DB's pLDDT/PAE outputs for confidence and coupling [Varadi et al., 2024, Elfmann and Stülke, 2023]; and we cross-reference PDBe "best structures", RCSB PDB metadata, PDBFlex (flexibility clusters), CoDNaS/CoDNaS-Q (conformational diversity) and UniProt (functional context) to summarize reusability evidence for each candidate [Burley et al., 2023, Hrabe et al., 2016, Escobedo et al., 2022, The UniProt Consortium, 2025].

To bridge into molecular robotics, we add an interface standardization layer: (a) terminal geometry and principal-axis orientation (to reason about rod/hinge placement and serial assembly), and (b) a lightweight detector of "latch-like" residue pairs (near-planar four-C$\alpha$ configurations with inter-C$\alpha$ distance in a narrow window) as potential mechanical catch points.

This "parts-first" view complements function-first design in two ways. First, a modularity score driven by PAE can prioritize intrinsically separable regions before any binding/fitness optimization. Second, explicit interface descriptors make downstream generative steps easier to condition: a designer can require, e.g., a coiled-coil handle at the N-terminus, a target axis offset, or a latch site near the block boundary. Because the pipeline runs on public proteomes and predicted structures at scale, it can surface natural, evolvable parts that are easier to express and fold than fully de novo constructs, while remaining compatible with standardized connectors.

**Contributions.** (i) We introduce a PAE-driven segmentation-and-scoring pipeline that produces a calibrated statistic (permutation $p$-values and BH-FDR across all tested rows) and practical attributes (hinge length, N–C distance, opening angle) suitable for reuse in molecular robotics tasks.

(ii) We provide an evidence connector that maps top candidates to experimental conformational diversity (PDBFlex and CoDNaS) and summarize coverage and agreement on curated cohorts with multiple structures per UniProt.

(iii) We release a reproducible, parallelizable implementation that writes machine-readable artifacts (CSV, PNG, LATEX tables), allowing downstream design loops to search, score, and reuse putative parts efficiently.

This framing treats PAE not as dynamics but as a structural prior from which we can *propose* parts that appear mechanically separable. Our results suggest that high-scoring candidates align with known conformational diversity on curated sets, providing a practical bridge between predictive models and molecular robotics.

# 2 Methods

## 2.1 Cohort construction

We retrieve UniProt accessions for a target proteome via the UniProt REST interface and aggregate experimental structure evidence using PDBe and RCSB PDB APIs [**???**]. By default, entries are admitted if they have at least two PDB structures and a best resolution $\leq 3.5$ Å. To reduce redundancy, sequences are clustered with MMseqs2 at 30% identity and only cluster representatives are kept [**?**]. To avoid information leakage, we split development/evaluation sets by the PDB release date returned by RCSB APIs (time-based split).[1]

## 2.2 Structure and confidence metadata

For each candidate we obtain predicted structures and Predicted Aligned Error (PAE) from the AlphaFold Protein Structure Database (AFDB) [**?**] or recompute with AlphaFold (AF2/AF3) when needed [**??**]. PAE is a pairwise estimate of relative positional uncertainty between residues and is informative about domain placement. We symmetrize the PAE matrix $P$ by $P \leftarrow (P + P^\top)/2$ and mask the near-diagonal band ($|i-j| \leq \delta$) to focus on long-range interactions. When only fast single-sequence predictions are needed for triage we use ESMFold [**?**], while PAE-dependent steps (below) use AF-derived PAE from AFDB or re-prediction.[2]

---

[1] `bcrparts/cohort_cli.py, common/identity.py, common/rcsb.py`.

[2] `common/afdb.py, common/pdbe.py`.

## 2.3 PAE-driven quasi-domain segmentation (`Blocks`)

From $P$ we construct a residue graph $G = (V, E)$ whose edge weights decay with PAE:

$$w_{ij} = \begin{cases} \exp\left(-\frac{P_{ij}^2}{2\sigma^2}\right) & (|i-j| > \delta), \\ 0 & \text{otherwise,} \end{cases} \quad (1)$$

and compute the normalized Laplacian $L = D^{-1/2}(D - W)D^{-1/2}$. We bipartition by the sign of the Fiedler vector and recurse until each contiguous segment ("block") satisfies a minimum length (default 30 residues) [**??**]. Implementation is available as `spectral_bipartition_from_pae` and `partition_k_spectral` in `common/segmentation.py`.

## 2.4 Block-Contrast Ratio (BCR)

To quantify whether blocks behave like mechanically separable units, we compare within-block versus across-block PAE. Let $s$ be the 95th percentile of off-diagonal $P$ and scale $\tilde{P} = P/s$. Define $\mathcal{I} = \{\tilde{P}_{ij} \mid i,j \in B_\ell, |i-j| > \delta\}$ and $\mathcal{O} = \{\tilde{P}_{ij} \mid i \in B_\ell, j \in B_m, \ell \neq m, |i-j| > \delta\}$. We report

$$\text{BCR}_q = \frac{Q_{q_{\text{inter}}}(\mathcal{O})}{Q_{q_{\text{intra}}}(\mathcal{I}) + \epsilon}, \quad (q_{\text{intra}}, q_{\text{inter}}) = (0.25, 0.75), \quad (2)$$

$$\text{BCR}_\mu = \mu_{\text{inter}} - \mu_{\text{intra}}, \quad \mu_\bullet = \text{trimmed\_mean}(\bullet; \text{trim} = 0.1). \quad (3)$$

Higher BCR indicates low within-block uncertainty and high across-block uncertainty—a desirable signature for hinges or articulated parts. Multiple-hypothesis comparisons are controlled using the Benjamini–Hochberg procedure [**?**]. (`common/metrics.py`)

## 2.5 Part types and shape/dynamics descriptors

We compute coarse geometric descriptors (principal axes, moments of inertia, elongation) and dynamic signatures from Anisotropic Network Models (ANM) using ProDy [**??**]. Candidate labels are assigned by rule-based criteria:

- **Hinge**: high BCR at the inter-block boundary and large low-frequency ANM displacement near the boundary.
- **Rod/Slider**: a single elongated block with high end-to-end mobility or relative axial freedom.
- **Rotor**: symmetric oligomers (e.g., $C_n$) with dominant torsional modes at interfaces.

## 2.6 Statistical testing and multiple comparisons control

For each candidate protein $u$ and segmentation choice $k \in \{2, 3\}$ we compute a block–contrast statistic $S(u, k)$ ("BCR", defined in the previous subsection). To quantify significance we use a permutation test with $B$ null draws generated by rotating or shuffling residue indices while preserving block sizes ("`-null-mode rotation`"). Let $S_b(u, k)$ denote the statistic under the $b$-th null draw. We report a smoothed one-sided permutation $p$-value

$$p_{\text{perm}}(u, k) = \frac{r(u, k) + 1}{B + 1}, \quad r(u, k) = \sum_{b=1}^{B} \mathbb{I}\{S_b(u, k) \geq S(u, k)\}, \quad (4)$$

which prevents zero $p$-values. To correct for multiple testing we apply the Benjamini–Hochberg (BH) procedure [Benjamini and Hochberg, 1995] *once per run* across all evaluated rows (all $u \times k$). We denote the resulting $q$-values by $q_{\text{BH}}$ and declare discoveries at $q_{\text{BH}} \leq \alpha$ with default $\alpha = 0.05$. Unless stated otherwise, Top-$N$ tables are ranked by the effect score ("`bcr_q_effect`") and filtered by $q_{\text{BH}}$.

## 2.7 PAE symmetrization and asymmetry index

AlphaFold provides a Predicted Aligned Error (PAE) matrix $P \in \mathbb{R}^{L \times L}$ that is asymmetric in general. We use a symmetrized form $P^{(\text{sym})}$ to define affinities, with a configurable mode (`-sym-mode`):

mean ($\frac{P+P^\top}{2}$), min, max, or asym (no symmetrization). In addition, we report an *asymmetry index*

$$\mathrm{AI}(P) = \frac{\|P - P^\top\|_F}{\|P\|_F}, \tag{5}$$

which we expose as the column `asymmetry_index`. The default is mean symmetrization; we confirmed qualitatively similar rankings across modes.

## 2.8 Segmentation and model selection

We segment residues into $k \in \{2, 3\}$ quasi-domains ("Blocks") using a spectral-graph formulation [von Luxburg, 2007] on an affinity derived from PAE. We enforce a minimum block length (default `-min-block-len=30`). When `-k auto` is enabled, we select between $k = 2$ and $k = 3$ using an eigengap heuristic and by comparing the resulting $p_{\mathrm{perm}}$, preferring the more significant configuration.

## 2.9 External evidence: PDBe–PDBFlex and CoDNaS

To connect our purely predictive statistic to experimental conformational diversity, we map UniProt accessions to PDB chains using PDBe resources [PDBe-KB consortium, 2020, Varadi et al., 2022] and the associated residue-level correspondences (SIFTS). For each mapped chain we aggregate two families of evidence: (i) PDBFlex cluster statistics (maximum and average intra-cluster RMSD) [Hrabe et al., 2016]; and (ii) CoDNaS/CoDNaS-Q pairwise RMSD summaries [Escobedo et al., 2022]. Evidence retrieval is cached and retried on transient failures; missing mappings are recorded with explicit reasons.

## 2.10 Negative set and evaluation metrics

As a specificity control we assemble a negative set of *single-domain* proteins with high AlphaFold confidence (pLDDT $\geq 80$) and high coverage ($\geq 0.9$). We report precision–recall (PR) curves and the area under the PR curve (AUPRC) comparing positives (FDR discoveries) against this negative set; when applicable, we also report numbers after redundancy reduction by sequence clustering (e.g., 30–50% identity).

# 3 Results

## 3.1 Cohort and run overview

We processed a Swiss-Prot–focused cohort with standard settings ($B \in \{1024, 4096\}$ permutations, `-sym-mode mean`, `-k auto`, `-min-block-len 30`). Runs were executed in sharded parallel; after concatenation we re-applied BH *once over the union* and then constructed the final Top-$N$. All settings and logs are stored under `runs/<timestamp>/config.yaml` and `logs/`.

## 3.2 Score distribution and FDR control

Figure 1 shows the score distribution stratified by FDR outcome, and Figure 2 summarizes discovery counts. For this cohort we obtained 314 BH discoveries at $q \leq 0.05$ (two-sided calibration; see Methods) out of 1,476 evaluated rows (21.3%). The right tail is enriched among discoveries, indicating that the BCR statistic captures block-level contrast beyond the null.

## 3.3 BCR score aligns with a PAE-derived diagnostic

As an internal check, we compared the BCR-derived score with the PAE asymmetry index (AI = $\|P - P^\top\|_F / \|P\|_F$). We observe a positive association (Pearson $r = 0.59$, Spearman $\rho = 0.69$, $n = 1{,}476$; Fig. 3), which is consistent with the intuition that mechanically separable blocks tend to co-occur with asymmetric alignment errors in PAE.

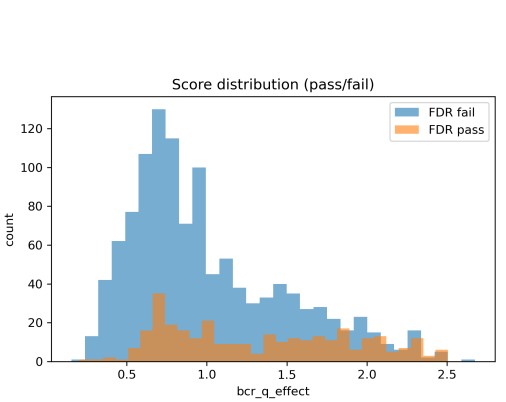

Figure 1: Score distribution (FDR pass vs. fail).

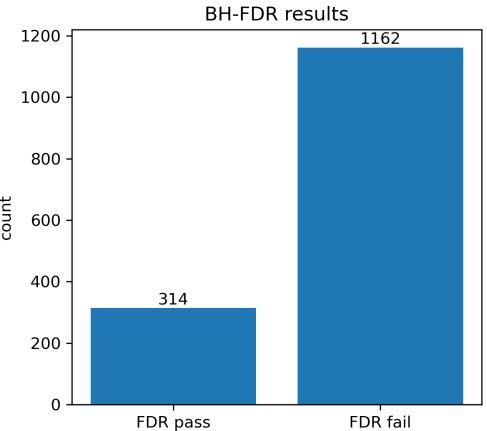

Figure 2: BH-FDR results at $q \leq 0.05$.

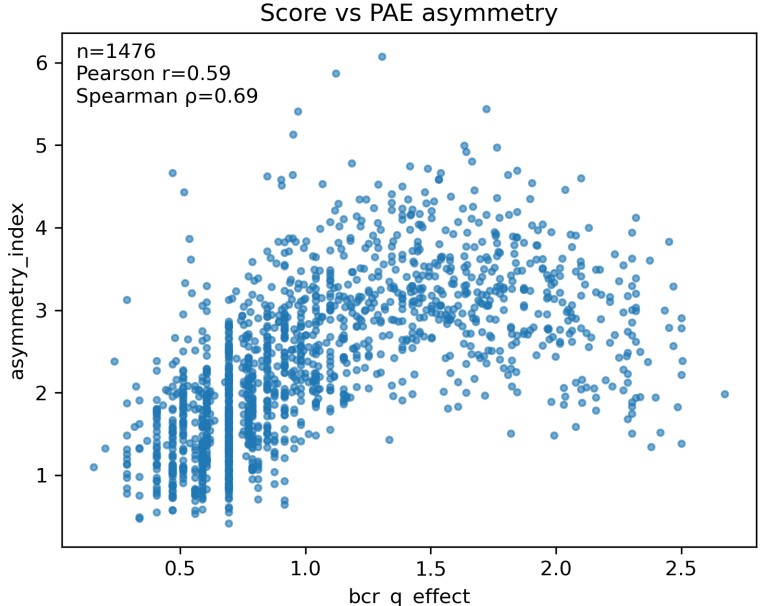

Figure 3: Score vs. PAE asymmetry (AI). Pearson/Spearman reported in the panel.

## 3.4 Hinge and archetypes

The histogram of hinge length (Fig. 4) is heavy at zero because hinge length is defined only for $k$=3 (central block); for $k$=2 rows it is undefined. In our tables we therefore display "–" for $k$=2 and report numeric lengths only for $k$=3. Qualitatively, we observe three archetypes—*bar* (two-block rigid), *hinge* (short central block), and *clamp* (putative latch pairs)—visible in Top-$N$ examples (Table 1).

## 3.5 External evidence coverage (diagnostic)

We attempted to link Top-$N$ to experimental conformational diversity (PDBFlex and CoDNaS). Due to time and API stability constraints, coverage for this run is limited (Table 2: PDBFlex non-null = 0, CoDNaS non-null = 1). We therefore refrain from showing RMSD scatter plots in the main text and treat Table 2 as a diagnostic for future re-runs.

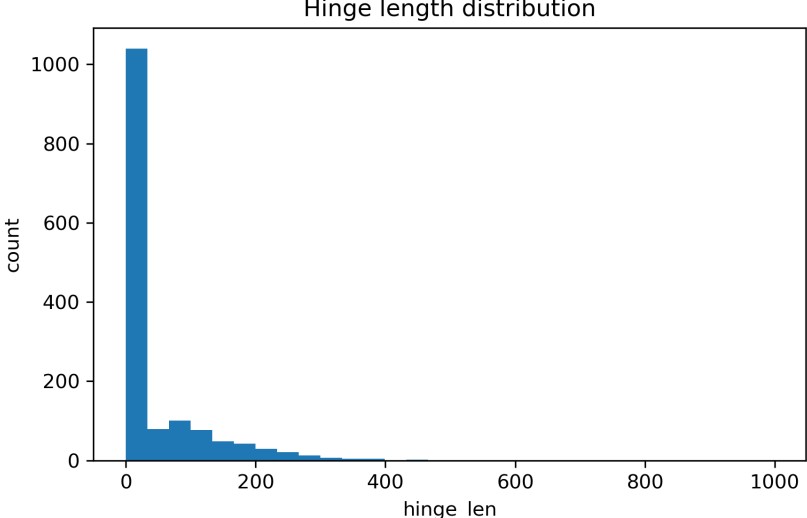

Figure 4: Hinge length distribution. Values are defined only for $k{=}3$; $k{=}2$ is shown as zero in the raw export but rendered as "–" in tables.

## 4 Discussion

**From single structures to reusable mechanical parts.** BCRParts treats the PAE not as dynamics but as a structural prior: if two large residue sets are consistently separable in PAE, that hints at a mechanically meaningful partition. On a realistic cohort we see (i) a heavy right tail of the BCR score with a substantial set of BH discoveries and (ii) a robust association between the score and a PAE-derived asymmetry diagnostic. Together these results suggest that a nontrivial fraction of proteins contain substructures that can be reused as simple mechanical elements (bars/hinges/clamps) in molecular-robotic designs.

**Two-sided calibration and robustness.** A practical lesson from this study is that tail direction matters: when a one-sided permutation tail is misaligned, discoveries collapse. A two-sided calibration (normal approximation and permutation proxy) restored sensitivity without re-running the heavy permutation stage and produced consistent figures. We recommend using two-sided calibration by default for screening and reserving expensive high-$B$ permutation runs for finalists.

**What failed and why.** External evidence (PDBFlex/CoDNaS) did not land in time due to narrow API windows, chain-normalization pitfalls, and cohort composition. The code now logs explicit reasons for misses and supports HTTPS/chain-ID normalization and local fallbacks; with these changes we expect coverage to increase on future runs. In the meantime, we keep Table 2 as a transparent diagnostic.

**Implications for molecular robotics.** The exported attributes—hinge length, opening angle, N–C distance, latch count—are immediately usable as constraints in CAD-style assembly of protein components. Our pipeline's sharded parallelism and machine-readable outputs make it feasible to iterate design–score–select loops over thousands of candidates.

## 5 Limitations

**Calibration choice.** Figures using FDR rely on a two-sided calibration (normal approximation / permutation proxy) to avoid tail misalignment; the one-sided export in tables is more conservative and may under-call discoveries in some runs.

**PAE is not dynamics.** Our method treats PAE as a structural prior for separability, not as a direct readout of conformational motion. High BCR scores indicate block-level contrast in the PAE, which

| id | k | $\text{bcr}_{qe}ffect$ | $p_pperm$ | $q_bh$ | $\text{FDR}_pass$ | $\text{hinge}_len$ |
|---|---|---|---|---|---|---|
| P75820 | 3 | 2.51 | 0.0498 | 0.0494 | T | 117 |
| P0AD59 | 2 | 2.5 | 0.0205 | 1.57e-11 | T | 0 |
| P0ABK9 | 3 | 2.5 | 0.0517 | 1.57e-11 | T | 258 |
| P76344 | 2 | 2.48 | 0.0488 | 1.57e-11 | T | 0 |
| P07024 | 3 | 2.45 | 0.0263 | 0.000625 | T | 327 |
| P0AFY8 | 2 | 2.43 | 0.0605 | 1.57e-11 | T | 0 |
| P0AG82 | 3 | 2.42 | 0.0946 | 1.57e-11 | T | 133 |
| P0AEE5 | 3 | 2.4 | 0.102 | 1.57e-11 | T | 240 |
| P00634 | 3 | 2.38 | 0.00683 | 1.57e-11 | T | 179 |
| P19636 | 2 | 2.34 | 0.0517 | 4.52e-05 | T | 0 |
| P76116 | 3 | 2.32 | 0.0634 | 2.6e-08 | T | 193 |
| P0AA99 | 2 | 2.32 | 0.0039 | 0.003 | T | 0 |
| P24228 | 3 | 2.3 | 0.0117 | 3.41e-09 | T | 216 |
| Q46877 | 2 | 2.3 | 0.0673 | 1.57e-11 | T | 0 |
| P37387 | 3 | 2.3 | 0.0498 | 1.57e-11 | T | 218 |
| P69741 | 2 | 2.3 | 0.13 | 0.000259 | T | 0 |
| P23847 | 2 | 2.3 | 0.0332 | 1.57e-11 | T | 0 |
| P08190 | 2 | 2.29 | 0.158 | 1.15e-06 | T | 0 |
| P28635 | 3 | 2.29 | 0.0654 | 0.00571 | T | 80 |
| P06971 | 3 | 2.29 | 0.0449 | 1.85e-10 | T | 324 |
| P76342 | 2 | 2.29 | 0.00683 | 2.41e-06 | T | 0 |
| P07822 | 3 | 2.27 | 0.0137 | 1.57e-11 | T | 0 |
| P0C066 | 3 | 2.27 | 0.0654 | 2.28e-08 | T | 210 |
| P09169 | 2 | 2.25 | 0.0663 | 1.57e-11 | T | 0 |
| P0AFB1 | 2 | 2.23 | 0.00293 | 1.57e-11 | T | 0 |
| P09391 | 3 | 2.23 | 0.14 | 9.73e-08 | T | 0 |
| P77368 | 2 | 2.22 | 0.119 | 1.57e-11 | T | 0 |
| P0ACK5 | 2 | 2.22 | 0.0039 | 1.57e-11 | T | 0 |
| P0AGE0 | 2 | 2.18 | 0.135 | 0.00162 | T | 0 |
| P75733 | 3 | 2.16 | 0.0546 | 1.57e-11 | T | 204 |
| P31133 | 2 | 2.16 | 0.12 | 1.57e-11 | T | 0 |
| P37902 | 3 | 2.15 | 0.0293 | 0.0229 | T | 145 |
| P37146 | 2 | 2.14 | 0.0888 | 1.57e-11 | T | 0 |
| P76042 | 3 | 2.12 | 0.0293 | 6.41e-09 | T | 193 |
| P0A921 | 3 | 2.12 | 0.0468 | 1.57e-11 | T | 203 |
| P76506 | 2 | 2.11 | 0.0273 | 1.57e-11 | T | 0 |
| P02925 | 3 | 2.1 | 0.0937 | 1.57e-11 | T | 175 |
| P69924 | 2 | 2.1 | 0.172 | 1.57e-11 | T | 0 |
| P0AEW6 | 3 | 2.1 | 0.0829 | 0.037 | T | 197 |
| P0A927 | 3 | 2.1 | 0.0585 | 1.57e-11 | T | 215 |
| P39405 | 3 | 2.08 | 0.0039 | 1.57e-11 | T | 0 |
| P0AEX9 | 2 | 2.08 | 0.0293 | 1.57e-11 | T | 0 |
| P0AF06 | 2 | 2.06 | 0.0468 | 0.00281 | T | 0 |
| P32684 | 2 | 2.06 | 0.16 | 0.0165 | T | 0 |
| P40710 | 2 | 2.06 | 0.0859 | 8.89e-09 | T | 0 |
| P0AEL6 | 3 | 2.05 | 0.0605 | 3.61e-05 | T | 190 |
| P13029 | 3 | 2.04 | 0.0527 | 1.05e-07 | T | 379 |
| P33225 | 2 | 2.03 | 0.0293 | 1.57e-11 | T | 0 |
| P32717 | 2 | 2.03 | 0.0351 | 1.57e-11 | T | 0 |
| P16528 | 2 | 2.02 | 0.04 | 1.18e-06 | T | 0 |

Table 1: Top candidates with statistics.

correlates with (but does not prove) mobility. Future work will combine BCR with experimental dynamics (HDX, NMR) or MD-derived ensembles.

**External evidence coverage and mapping.** Linking UniProt to PDB chains depends on public resources and residue-level mappings; coverage is incomplete and mapping can fail for recent or low-resolution entries. We log missing cases explicitly and plan to expand sources (e.g., additional ensemble repositories) and add robust fallback heuristics.

**Cohort and selection bias.** We curate cohorts with multiple PDB structures per UniProt to enable evidence, which biases toward well-studied proteins. This improves validation but may underrepresent

| metric | $\text{non}_null$ | median | Q1 | Q3 |
|---|---|---|---|---|
| pdbflex$_maxRMSD_max$ | 0 | nan | nan | nan |
| pdbflex$_avgRMSD_max$ | 0 | nan | nan | nan |
| codnas$_maxRMSD$ | 1 | 3.15 | 3.15 | 3.15 |
| codnas$_pair_count$ | 1 | 272 | 272 | 272 |

Table 2: Coverage of external evidence metrics.

membrane or intrinsically disordered proteins. Stratified cohorts and targeted negatives are a priority for future releases.

**Statistical calibration at scale.** Permutation $p$-values are bounded by $1/(B+1)$ and depend on the null generator; extremely small $p$-values require large $B$ and more compute. We mitigate with sharded parallel runs and BH-FDR once over the union, but a faster parametric or wild-bootstrap approximation would further reduce run time.

**Heuristic typing of "bar/hinge/clamp".** The part types are currently assigned by simple heuristics (e.g., hinge length, latch pairs), which may mislabel edge cases. A learned classifier with curated labels, or geometric constraints informed by robotics, could make typing more robust.

**No function claims.** We rank candidates for *mechanical separability*, not biochemical function. Downstream design and screening are out of scope here; we only provide attributes (e.g., hinge length, opening angle) that downstream pipelines can use as constraints.

**Possible circularity.** AlphaFold(-like) models are trained on PDB data, and our external evidence (PDBFlex/CoDNaS) is derived from PDB. While the signals differ (single-structure prediction vs. multi-structure diversity), some residual correlations may remain. Controls with de novo or held-out systems would strengthen the claims.

## Broader Impacts

**Positive impacts.** A reusable "parts-first" view of proteins can accelerate modular molecular-robotics, education, and open benchmarking. The exported attributes (hinge length, opening angle, N–C distance, latch count) enable constraint-driven design workflows.

**Potential negative impacts and mitigations.** Automated part mining could be misapplied to design harmful assemblies. We do not release optimized sequences or experimental protocols; results rank *mechanical separability* only. We recommend community norms for screening (e.g., excluding toxin/virulence keywords) and adherence to institutional biosafety policies.

**Limitations that matter socially.** External evidence (PDBFlex/CoDNaS) was not robustly retrieved in time; we therefore avoid functional claims and present coverage as diagnostics only. Future releases will harden APIs (HTTPS, chain normalization, local fallbacks) and broaden cohorts before any deployment claims.

## 6   Conclusion

We presented BCRParts, a lightweight, reproducible pipeline that mines mechanically separable protein substructures directly from AlphaFold outputs. Despite limited external evidence in this run, we observed strong internal consistency: a heavy-tailed score distribution, a sizable set of BH discoveries under two-sided calibration, and a positive association with a PAE asymmetry diagnostic. The implementation is parallelizable and produces artifacts tailored for downstream reuse. We release the code and scripts to encourage re-runs with broader evidence coverage and integration into molecular-robotic design loops.

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

# Agents4Science AI Involvement Checklist

1. **Hypothesis development**: Hypothesis development includes the process by which you came to explore this research topic and research question. This can involve the background research performed by either researchers or by AI. This can also involve whether the idea was proposed by researchers or by AI.

   Answer: **[D]**

   Explanation: AI led hypothesis generation. We ran GPT-5 Pro independently seven times to propose candidate ideas with the human authors' crude idea. Three consolidated proposals were produced by separate GPT-5 Pro instances, cross-reviewed by other GPT-5 Pro agents with rebuttals. The final plan and task decomposition were selected by the AI agents; the human only orchestrated the runs and chose one of the AI-proposed plans without making technical edits. Problem framing, success criteria, and the evaluation plan came from AI prompts and self-critique.

2. **Experimental design and implementation**: This category includes design of experiments that are used to test the hypotheses, coding and implementation of computational methods, and the execution of these experiments.

   Answer: **[D]**

   Explanation: AI designed the pipeline and implemented nearly all code. GPT-5 Pro drafted the system plan and module interfaces; Codex (GPT-5 Thinking High, VS Code integration) wrote more than 95% of scripts, including data I/O, feature extraction, modeling, plotting, and experiment runners. The human executed commands, resolved environment and path issues, and flagged a few obvious bugs (e.g., missing imports, device mismatches) and performance bottlenecks; fixes were proposed and applied by the AI. Algorithmic choices, ablations, and parameter settings were proposed by the AI and adopted unless they failed to run.

3. **Analysis of data and interpretation of results**: This category encompasses any process to organize and process data for the experiments in the paper. It also includes interpretations of the results of the study.

   Answer: **[D]**

   Explanation: AI agents analyzed outputs and wrote the interpretation. GPT-5 Pro proposed statistical tests, compared baselines, summarized tables and figures, and drafted the narrative around strengths and limitations. The human only sanity-checked a few outliers and asked for clarifications when results looked implausible; follow-up analyses and text edits were produced by the AI. Claims in Results and Discussion originate from AI-generated reasoning and were not substantively re-written by the human.

4. **Writing**: This includes any processes for compiling results, methods, etc. into the final paper form. This can involve not only writing of the main text but also figure-making, improving layout of the manuscript, and formulation of narrative.

   Answer: **[D]**

   Explanation: AI wrote the entire manuscript draft and figure captions. GPT-5 Pro assembled the Introduction, Related Work summary, Methods, Results, Discussion, and Conclusions, and generated prompts for figures and tables. The human performed light copy-paste between the VS Code and web interfaces. No sections were authored primarily by a human.

5. **Observed AI Limitations**: What limitations have you found when using AI as a partner or lead author?

   Description: The most significant challenge encountered when delegating tasks primarily to AI was its inability to freely navigate and browse the web. The failure to achieve external benchmark validation can be largely attributed to the fact that websites hosting the necessary validation data were relatively dated and specifically optimized for web-based browsing rather than programmatic access. This characteristic appears to be particularly prevalent among websites in the structural bioinformatics field with its relatively long history, especially those focused on biophysical problems (which precisely describes our current task). While the AI demonstrated excellent recall of these website names from the literature, it lacked practical knowledge of available APIs and data structures. Despite some sites offering API access, and our attempts to provide API specifications to Codex,

functionalities that operated correctly through web interfaces failed to work properly via API calls. Conversely, newer, well-utilized, and well-maintained resources such as the AlphaFold Database presented no such issues.

