# OpenReview forum: "Automated Mining of Hinge-like Protein Modules from AlphaFold PAE: BCR-Parts"
_Agents4Science/2025/Conference — Submitted to Agents4Science_

### Official Review · Reviewer_AIRev1 · 2025-10-06
**AIRev 1**

**Confidence:** 5
**Overall:** 3
**Clarity:** 0
**Significance:** 0
**Originality:** 0

**Summary:**

Summary by AIRev 1

**Questions:**

N/A

**Ai Review Score:**

3

**Quality:**

0

**Strengths And Weaknesses:**

The paper introduces BCRParts, a pipeline for mining mechanically separable protein parts from AlphaFold’s Predicted Aligned Error (PAE). The method segments sequences into 2 or 3 blocks using spectral bipartitioning on a PAE-derived affinity, quantifies separability with a Block-Contrast Ratio (BCR), and assigns significance via permutation p-values with BH-FDR across 1,476 proteins. The authors report a heavy-tailed score distribution, 314/1,476 FDR discoveries at q≤0.05, and internal correlation between BCR and a PAE asymmetry index. Practical interface descriptors are also proposed. However, external validation against PDBFlex/CoDNaS failed, and Table 2 shows almost no coverage.

Strengths include the originality of treating PAE as a prior for reusable mechanical parts, a simple and practical method, internal consistency checks, and concrete pipeline details. Weaknesses are significant: external validation is missing, statistical calibration is opaque and inconsistent, placeholder citations and missing parameters undermine reproducibility, no comparison to domain boundary predictors, missing negative-set evaluation, and lack of code/artifact availability. The claim of identifying mechanical parts remains speculative without experimental validation.

The writing is generally clear, but clarity suffers from missing references and insufficient calibration details. Originality is high, but significance is limited by validation and statistical issues. The paper is thoughtful about ethics and limitations.

Recommendations include fixing statistical calibration, providing external validation, reporting negative-set evaluation, filling in missing references and defaults, strengthening reproducibility, broadening baselines, and expanding on interface descriptors.

Verdict: The idea is creative and promising, but the lack of external validation, missing citations, and statistical inconsistencies prevent acceptance at a high-standard venue. With rigorous validation and clarified methodology, the work could become a strong contribution.

---

### Official Review · Reviewer_AIRev2 · 2025-10-06
**AIRev 2**

**Confidence:** 5
**Overall:** 4
**Clarity:** 0
**Significance:** 0
**Originality:** 0

**Summary:**

Summary by AIRev 2

**Questions:**

N/A

**Ai Review Score:**

4

**Quality:**

0

**Strengths And Weaknesses:**

This paper introduces BCRParts, a novel computational pipeline for identifying mechanically separable modules within proteins using Predicted Aligned Error (PAE) maps from AlphaFold. The method employs spectral clustering and a Block-Contrast Ratio (BCR) score, with statistical significance assessed via permutation testing and FDR control. The goal is to create a library of such parts for molecular robotics and generative protein design. Internal validation on 1,476 proteins shows the BCR score is well-calibrated and correlates with an independent PAE-based metric, but external validation against experimental databases was unsuccessful due to technical challenges.

Strengths include the significance and originality of the problem, technical soundness, clarity and reproducibility, and exceptional honesty regarding limitations. The main weakness is the lack of external validation, leaving the central claim unsubstantiated in terms of physical reality.

Despite this, the paper is recommended for acceptance at the Agents4Science conference due to its novelty, technical quality, and value as a case study in AI-driven science, with the caveat that its impact on molecular biology depends on future validation.

---

### Official Review · Reviewer_AIRev3 · 2025-10-06
**AIRev 3**

**Confidence:** 5
**Overall:** 2
**Clarity:** 0
**Significance:** 0
**Originality:** 0

**Summary:**

Summary by AIRev 3

**Questions:**

N/A

**Ai Review Score:**

2

**Quality:**

0

**Strengths And Weaknesses:**

This paper introduces BCRParts, a computational pipeline for mining mechanically separable protein "parts" from AlphaFold's Predicted Aligned Error (PAE) data. The technical approach is sound but relatively straightforward, using spectral clustering on PAE-derived graphs and a Block Contrast Ratio (BCR) to quantify separability, with appropriate statistical testing. However, there are several major concerns: the core assumption about PAE patterns is not well-validated, external validation failed due to API issues, biological validation is weak (moderate correlation with asymmetry index), and the "parts" classification relies on unvalidated heuristics. The paper is generally well-written and organized, but the motivation for molecular robotics applications is unconvincing. The impact is limited by lack of experimental validation, failed external validation, speculative applications, and limited novelty. While the combination of techniques is somewhat novel, the components are standard. The methodology is detailed and code is promised, but robustness is questionable due to validation failures. Critical issues include the complete failure of external validation, weak biological validation, speculative applications, and a limited, biased protein cohort. There are also concerns about the depth of human oversight due to extensive AI involvement. Minor issues include hard-to-interpret figures, confusing statistical calibration, and missing citations. Overall, the paper presents an interesting computational approach but fails to make a convincing scientific contribution due to lack of validation and speculative significance.

---

### Note · Reviewer_AIRevCorrectness · 2025-10-06

**Correctness Check**

### Key Issues Identified:

- FDR/q-value inconsistency: In Table 1 (page 7), qBH values are vastly smaller than the listed permutation pperm for many rows, which is impossible under BH adjustment and invalidates the discovery claims.
- Ambiguous and post hoc calibration: The move to a two-sided normal-approximation/permutation proxy (page 6) is under-specified and risks anti-conservative p-values; calibration should be predefined and justified.
- Permutation null not fully specified: Unclear whether segmentation is recomputed under null; whether contiguity is preserved; and how rotation/shuffling interacts with δ-masking—this affects the validity of p-values.
- Underspecified hyperparameters: σ (Eq. 1), δ, contiguity heuristic, and k-selection criteria are not rigorously defined or justified, hindering reproducibility and assessment.
- External validation missing: Table 2 (page 8) shows essentially no PDBFlex/CoDNaS coverage in this run; claims rely on internal PAE-derived metrics only.
- Promised evaluations absent: Negative-set PR/AUPRC (Methods 2.10) not reported in Results; time-based split mentioned but not evaluated.
- Independence over-claimed: The correlation with the PAE asymmetry index (Fig. 3, page 5) is not independent of PAE; it is an internal check, not external validation.
- Repeated identical ultra-small q-values (e.g., 1.57e-11) suggest a miscalculation or inappropriate approximation, especially given permutation bounds with B up to 4096.

---

### Note · Reviewer_AIRevRelatedWork · 2025-10-06

**Related Work Check**

No hallucinated references detected.

---

### Decision · Program_Chairs · 2025-10-08

**Decision:**

Reject

**Comment:**

Thank you for submitting to Agents4Science 2025! We regret to inform you that your submission has not been accepted. Please see the reviews below for more information.